SARS-CoV-2 seroprevalence in a high-altitude setting in Peru: adult population-based cross-sectional study

Huamaní Charles chuamani@uandina.edu.pe 1 2
Velásquez Lucio 1 2
Montes Sonia 2
Mayanga-Herrera Ana 3
Bernabé-Ortiz Antonio 3
1 Universidad Andina del Cusco , Cusco , Peru
2 Hospital Nacional Adolfo Guevara Velasco , Cusco , Peru
3 Universidad Científica del Sur , Lima , Peru
Palazón-Bru Antonio
Electronic publication date: 2021 Sep 20
Publication date: 2021
Volume: 9
Electronic Location ID: e12149
Received 2021 Mar 11; Accepted 2021 Aug 20
Copyright: ©2021 Huamani et al.
Copyright year: 2021
Copyright holder: Huamani et al.
License: This is an open access article distributed under the terms of the Creative Commons Attribution License, which permits unrestricted use, distribution, reproduction and adaptation in any medium and for any purpose provided that it is properly attributed. For attribution, the original author(s), title, publication source (PeerJ) and either DOI or URL of the article must be cited.
License URL: https://creativecommons.org/licenses/by/4.0/

Keywords: COVID, Altitude, Peru, Seroprevalence

Funding: FONDECYT (Fondo Nacional de Ciencia y Tecnología, Peru) 071-2020 This study was supported by FONDECYT (Fondo Nacional de Ciencia y Tecnología, Peru), code: 071-2020. The funders had no role in study design, data collection and analysis, decision to publish, or preparation of the manuscript.

==============================
Background

There are several ecological studies, but few studies of the prevalence of SARS-COV-2 at high altitude. We aimed to estimate the population-based seroprevalence of SARS-COV-2 in three settings of Cusco at the end of the first wave among adults.

Methods

A population-based survey was conducted in September 2020, in three settings in the region of Cusco: (1) Cusco city at 3,300 meters above the sea level (m.a.s.l.), (2) the periphery of Cusco (Santiago, San Jerónimo, San Sebastián, and Wanchaq) at 3,300 m.a.s.l., and (3) Quillabamba city, located at 1,050 m.a.s.l. People aged ≥ 18 years within a family unit were included. The diagnosis of SARS-CoV-2 infection was based on identifying anti- SARS-CoV-2 total antibodies (IgM and IgG) in serum using the Elecsys Anti-SARS-CoV-2 chemiluminescence test.

Results

We enrolled 1924 participants from 712 families. Of the total, 637 participants were anti-SARS-CoV-2 seropositive. Seroprevalence was 38.8% (95% CI [33.4%–44.9%]) in Cusco city, 34.9% (95% CI [30.4%–40.1%]) in the periphery of Cusco, and 20.3% (95% CI [16.2%–25.6%]) in Quillabamba. In 141 families (19.8%; 95% CI [17.0%–22.8%]) the whole members were positive to the test. Living with more than three persons in the same house, a positive COVID-19 case at home, and a member who died in the last five months were factors associated with SARS-COV-2 seropositivity. Dysgeusia/dysosmia was the symptom most associated with seropositivity (aPR = 2.74, 95% CI [2.41–3.12]); whereas always wearing a face shield (aPR = 0. 73; 95% CI [0.60–0.89]) or a facial mask (aPR = 0.76, 95% CI [0.63–0. 92) reduced that probability.

Conclusions

A great proportion of Cusco’s city inhabitants presented anti-SARS-CoV-2 antibodies at the end of the first wave, with significant differences between settings. Wearing masks and face shields were associated with lower rate of seropositivity; however, efforts must be made to sustain them over time since there is still a high proportion of susceptible people.

Introduction

Worldwide, the Coronavirus Disease 19 (COVID-19) pandemic, caused by SARS-CoV-2 virus, has been evaluated in real-time through the official notifications of the affected countries (World Health Organization, 2020b), which usually come from passive surveillance systems. However, a great proportion of individuals infected by SARS-CoV-2 virus remains undetected, especially in resource-constrained settings, as they usually are asymptomatic (Gandhi, Yokoe & Havlir, 2020), and there is a lack of appropriate access to diagnosis in the health care system, generating a gap in the information for appropriate decisions (Byambasuren et al., 2020; Eckerle & Meyer, 2020). To deal with these issues, multiple population-based surveys have been conducted around the world (Franceschi et al., 2020; Lai, Wang & Hsueh, 2020; Rostami et al., 2020), with divergent results as countries are in different epidemiological scenarios, i.e., beginning or end of the first pandemic wave, urban/rural areas, national/regional representation, or different diagnostic test used (molecular, antibody, or antigen detection tests). Even so, reported prevalence has usually been lower than 20% after the end of the first wave (Rostami et al., 2020).

Few studies have been published in low- and middle-income countries, including Latin American countries (Lai, Wang & Hsueh, 2020; Rostami et al., 2020), where because different social determinants (e.g., poverty levels, inequities, overcrowding, and a weak health system), a higher anti-SARS-CoV-2 antibodies prevalence could be expected (Burki, 2020; Pablos-Méndez et al., 2020). Moreover, many of the studies in high-altitude cities (i.e., those located over 2,500 m above the sea level (m.a.s.l.)) (Arias-Reyes et al., 2020; Cano-Pérez et al., 2020; Segovia-Juarez, Castagnetto & Gonzales, 2020; Thomson et al., 2021; Woolcott & Bergman, 2020) are ecological in nature. Ecological studies include only registered cases; thus, the percentage of underreporting is high due to asymptomatic forms, diagnostic deficiencies, limitations to access health services, among others. These limitations are not present in studies like ours, as they are based on probabilistic samples and not only in the presence of symptoms. Also, many ecological studies have suggested that the impact of the COVID-19 pandemic would be low because unclear environmental determinants such as atmospheric pressure or radiation (Segovia-Juarez, Castagnetto & Gonzales, 2020; Thomson et al., 2021). However, the pandemic progression is more linked to social interaction and adopted preventive measures (Huamaní et al., 2020).

By September 2020, Peru had registered more than 900 thousand confirmed cases and more than 32 thousand deaths attributed to the pandemic (Ministerio de Salud, 2021). For that reason, Peru has been on the top of number deaths by COVID-19 per million of people in South America (World Health Organization, 2020b). The Peruvian government response to the pandemic was prompt, including mandatory social distancing measures, mandatory use of masks, and re-focusing most health system resources to address the pandemic. Peru is a very heterogeneous country with a variable geography, thus, although the first case of SARS-COV-2 was detected in March 2020, some high-altitude cities like Cusco experienced the first wave between July and September (Dirección Regional de Salud de Cusco, 2020). At the end of this period, we conducted a population-based seroprevalence of anti-SARS-CoV-2 antibodies in different areas of Cusco, and evaluated some determinants associated with the spread of the SARS-CoV-2 seropositivity.

Materials & Methods

Study design

A population-based cross-sectional study was conducted between September 12 and 27, 2020 in the region of Cusco, Peru.

Study locations

The region of Cusco comprises thirteen provinces, each having a different number of cities (Fig. 1). Cities offering variability of scenarios were chosen due to their proportion of urban/rural areas, altitude, and population. Three different study settings were selected as follows:

Figure 1 Map of Cusco, provinces, and cities.

Cusco has thirteen provinces (middle image). The capital of La Convención Province is Quillabamba (yellow dot), located at 1,050 m.a.s.l. The Cusco Province (blue arrow), at 3,300 m.a.s.l., is inhabited in only one sector located in the valleys (blue mark in A), which include the five cities where the research was carried out (B).

- Cusco city, Province of Cusco, with a high demographic density (approximately 1,000 inhabitants/km2), population (125,000 inhabitants), and located at 3,330 m.a.s.l. Based on the information available about the behavior of the COVID-19 pandemic in the region (Dirección Regional de Salud de Cusco, 2020), it would represent the worst transmission scenario.

- Periphery of Cusco, Province of Cusco, including different cities around the Cusco city (Santiago, San Sebastián, San Jerónimo, and Wanchaq), where approximately 320,000 inhabitants in total live (approximately 1,100 inhabitants/km2). Although these cities are at the same altitude as Cusco city, they are located on the historic center’s periphery.

- Quillabamba City, Province of La Convención, located at the lowest altitude in the department of Cusco (1,050 m.a.s.l.) with an approximate population of 30,000 inhabitants, and geographically distant (approximately 6 h by road, not accessible by any other way), with a low population density (<10 inhabitants/km2). This city is part of the Peruvian jungle and, as a result, the climate and geography are different from that of Cusco city. Despite of that, during the first wave, La Convención was the second Province of Cusco with the most reported cases (Dirección Regional de Salud de Cusco, 2020).

Participants and sampling

People aged ≥18 years old, who voluntarily agreed to participate in the study, signed their informed consent, accepted the telephone and serological follow-up, usual resident of the study area (≥6 months), and with the ability to understand the procedures were included. We did not exclude people with acute symptoms or who had already given positive results to previous tests.

A two-stage probability sampling approach was carried out: by clusters of groups of blocks and by households. Maps by each of the cities were used as sampling frame. The primary sampling unit was defined as the cluster comprised by a block or group of blocks with approximately 40 households. In the Cusco city, 98 clusters were created, whereas these numbers were 495 and 74 for the periphery of Cusco and Quillabamba, respectively. Of them, 10, 16 and 8 clusters were randomly chosen in each of the settings, respectively. Within each cluster, households were selected randomly. In each selected household, all the members who met the eligibility criteria were included until the intended sample size was achieved.

The sample size calculation was based on an expected population prevalence of at least 5%, precision of 2.5%, a confidence level of 95%, and a design effect of 2. Based on these estimates, the required sample size in a conservative setting was 1752 participants for the three settings. With this sample size, we had a power over 80% to detect a difference in the prevalence of SARS-CoV-2 seropositivity of at least 5% (e.g., 5% vs. 10%) between the groups of interest (study setting and gender). Therefore, at least 1800 participants had to be enrolled for the three settings, considering losses to follow-up, and split into 600 participants for Cusco city, 800 for the periphery of Cusco, and 400 for Quillabamba.

Procedures

The collection of information and biological samples was carried out during three weekends in September 2020 (12, 13, 19, 20, 26 and 27) taking advantage of the period of lockdown in the region. Data and sample collection were prioritized during Sundays to guarantee the presence of the largest number of family members. The field staff visited each household to contact potential participants, assess eligibility, invite them to participate in the study, apply the informed consent, questionnaires, and finally, take blood samples.

The questionnaire evaluated sociodemographic characteristics (age, gender, etc.), comorbidities (self-report of hypertension, diabetes, asthma, cancer, renal or cardiac disease), weight and height (self-reported and with that, the BMI was calculated), obesity (BMI ≥ 30), number of people living together, past case at home of a patient with COVID-19, a family member who died in the last five months, among others. We included questions about symptoms developed in the last three months, self-report of any previous SARS-COV-2 diagnostic test (rapid or molecular test) performed in the last three months (tests are periodically done in some populations as part of the government indications. Thus, with this question we tried to know if the test of the survey was the first of the participant), and protective behaviors used or applied during quarantine (when leaving home, you have always used…a mask, alcohol for hand disinfection, gloves, etc.), including those mandatories for use by the government, but not regulated or sanctioned, such as the use of masks.

Once the questionnaire was completed, a blood sample of 3.5 mL of whole blood was taken in serum separating tubes. Each tube was coded and stored for transportation. The samples were stored between 2 to 8 °C for up to 5 days, according to the World Health Organization guidelines (World Health Organization, 2020a).

Chemiluminescence tests were used for serological detection of antibodies against SARS-CoV-2 Elecsys from the ROCHE laboratory. This test is based on a sandwich-type immunoassay, where the recombinant protein N of SARS-CoV-2 is the target detected by possible antibodies present in the serum sample (ROCHE S.A.C, 2020). The test detects total antibodies simultaneously, without differentiating between IgM or IgG. This test has a specificity of 99.5%, and whether the test is performed after 14 days of having a positive result by PCR, the sensitivity may reach up to 99.8% (Muench et al., 2020). In an evaluation carried out at the Instituto Nacional de Salud National Institute of Health (NIH) of Peru, it was confirmed that the sensitivity was 96.7% and the specificity was 98.6% (Instituto Nacional de Salud of Peru. Lima, 2020).

Statistical analysis

For data analysis, STATA 16 for Windows (StataCorp, College Station, TX, US) was utilized. The description of the study population was carried out according to the characteristics of interest. The prevalence of serum antibodies against SARS-CoV-2 virus (percentage of people seropositive to SARS-CoV-2) was estimated, taking into account the sampling techniques used. Prevalence estimates and regression analysis were calculated using Poisson distribution models adjusting for clusters at household level and with robust variance. Regression models were used to identify the factors associated with positivity for SARS-COV-2, obtaining adjusted prevalence ratios (aPR) for study settings, gender, and age group (<40, 40–59, and 60 + years of age), considering clusters at the household level. No other variables were included in the adjusted model because they did not show changes in the results and an attempt was made to preserve the parsimony of the model. All estimates are presented with 95% confidence intervals (95%CI).

Ethics

The study was approved by the Ethics Committee of the Universidad Científica del Sur (code 051-2020-PRO99). Written informed consent was used, and although personal identifiers were collected, this was done to guarantee appropriate delivery of test results. All the results were delivered directly to study participants. To reduce the risks of contagion by SARS-CoV-2 virus during the execution of the research project, all the interviewers followed the protocols for handling COVID-19 patients and providing masks to study participants. The participants had access to help lines, where trained physicians answered their questions and provided guidance concerning COVID-19.

Results

Characteristics of the study population

A total of 712 families (range: 1 to 11 members included) were enrolled in the present study, with a total of 1924 participants being evaluated. Of them, 408 (21.2%) were from Quillabamba, 640 (33.3%) were from Cusco city, and 876 (45.5%) were from the periphery of Cusco. The average age was 42.5 (SD: 16.5), and 1096 (57.1%) were women; these characteristics were not different between the different study settings. However, self-reported obesity was more frequent (26.5%) in Quillabamba compared to Cusco city (13.2%) or the periphery of Cusco (18.3%, p-value < 0.001). The characteristics of the study population, according to the study settings, are shown in Table 1.

Table 1 Population characteristics according to the study settings.

	Total	Cusco city	Periphery of Cusco	Quillabamba	p-value	
	(n = 1924)	(n = 640)	(n = 876)	(n = 408)		
Gender						
Female	1096 (57.1%)	359 (56.3%)	500 (57.1%)	237 (58.2%)	0.822	
Age: Mean (S.D.)	42.47 (16.5)	42.8 (16.2)	42.5 (16.8)	42.81 (16.5)	0.884	
Age group (years)						
≥ 18, <40	876 (46.5%)	298 (47.3%)	403 (46.8%)	175 (44.4%)	0.843	
40–59	702 (37.2%)	235 (37.3%)	313 (36.3%)	154 (39.1%)		
>60	308 (16.3%)	97 (15.4%)	146 (16.9%)	65 (16.5%)		
People living in the same house: Mean (SD)	5.98 (3.43)	6.24 (3.40)	6.16 (3.58)	5.04 (2.92)	<0.001	
Previous SARS-COV-2 test (rapid test or molecular)	758 (39.7%)	242 (38.1%)	328 (37.7%)	188 (46.7%)	0.006	
Obesity	332 (18.4%)	86 (13.2%)	152 (18.3%)	94 (26.5%)	<0.001	
No symptoms in the last 3 months	772 (40.2%)	236 (36.9%)	300 (34.4%)	236 (57.9%)	<0.001	
Self-reported comorbidities						
Hypertension	128 (6.7%)	43 (6.7%)	55 (6.3%)	30 (7.4%)	0.763	
Diabetes	85 (4.4%)	19 (3.0%)	37 (4.2%)	29 (7.1%)	0.006	
Asthma	29 (1.5%)	10 (1.6%)	13 (1.5%)	6 (1.5%)	0.990	
Cardiac disease	12 (0.6%)	5 (0.8%)	6 (0.7%)	1 (0.3%)	0.543	
Renal disease	19 (1.0%)	5 (0.8%)	12 (1.4%)	2 (0.5%)	0.271	
Cancer	10 (0.5%)	4 (0.6%)	4 (0.5%)	2 (0.5%)	0.900	

Overall, 39.7% (95% CI [36.8%–42.9%]) of participants indicated at least one previous test to detect SARS-COV-2. Being asymptomatic (aPR = 0.79, 95% CI [0.69–0. 91]) or being a woman (aPR = 0.82, 95% CI [0.74–0.91]) reducing the probability of having a previous test; whereas higher education (>12 years) increased such probability (aPR = 1.32, 95% CI [1.23–2.13]).

From 606 families that reported information, 46 (7.6%) stated that at least one member died in the 5 months previous to the survey, and from these, 60.5% attributed the cause of death to COVID-19.

Prevalence of anti-SARS-CoV-2 antibodies

A total of 637 participants were reactive to the screening test, which defines an adjusted prevalence of 33.1% (95% CI [30.1%–36.4%]). This prevalence varied according to the study settings: 20.3% (95% CI [16.2%–25.6%]) in Quillabamba, 38.8% (95% CI [33.4%–44.9%]) in Cusco city, and 34.9% (95%CI: 30.4%-40.1%) in the periphery of Cusco (Table 2, extended analysis on supplementary table).

Table 2 Prevalence of serum antibodies to SARS-CoV-2 in the general population at different altitudes in Cusco, Peru, according to individual/family characteristics.

Models adjusted for age, gender, and study setting.

	% seropositive to SARS-CoV-2 antibodies	Adjusted PR	
Global prevalence	33.1% (30.1–36.4%)		
Gender			
Male	31.1% (27.3–35.2%)	1	
Female	34.6% (31.2–38.3%)	1.09 (0.97–1.24)	
Study setting			
Cusco city	38.8% (33.4–44.9%)	1.85 (1.41–2.43)	
Periphery of Cusco	34.9% (30.4–40.1%)	1.71 (1.31–2.22)	
Quillabamba	20.3% (16.2–25.6%)	1	
Age group (years)			
≥ 18, <40	34.8% (31.0–39.1%)	1	
40–59	35.8% (31.8–40.2%)	0.99 (0.86–1.14)	
>60	24.7% (19.7–30.9%)	0.65 (0.51–0.83)	
Education level (years)			
<7	34.2% (26.9–43.3%)	1	
7–11	39.2% (34.4–44.7%)	1.09 (0.86–1.40)	
>12	30.3% (26.9–34.0%)	0.79 (0.61–1.03)	
BMI			
Normal	34.1% (30.0–38.6%)	1	
Overweight	31.1% (27.1–35.7%)	0.95 (0.81–1.13)	
Obese	35.5% (30.2–41.8%)	1.17 (0.95–1.44)	
Number of people living together			
1–2	20.5% (14.4–29.1%)	1	
3–5	31.8% (27.6–36.7%)	1.47 (1.01–2.15)	
6–10	37.5% (31.9–44.0%)	1.68 (1.14–2.46)	
>11	38.7% (28.1–53.2%)	1.71 (1.07–2.73)	
Self-report of COVID-19 case at home			
No	26.2% (22.9–29.9%)	1	
Yes	50.2% (44.4–56.7%)	1.90 (1.60–2.27)	
Death at home (any cause)			
No	31.9% (28.7–35.3%)	1	
Yes	50.8% (38.4–67.2%)	1.50 (1.11–2.04)	
Previous SARS-COV-2 test (rapid test or molecular)			
No	28.7% (25.3–32.6%)	1	
Yes	39.9% (35.7–44.7%)	1.46 (1.26–1.69)	

Of the 712 families evaluated, 318 (44.6%; 95% CI [41.0%–48.3%]) had at least one infected member, and 141 (19.8%; 95% CI [17.0%–22.8%]) families had the whole members positive to the test.

Factors associated with SARS-CoV-2 seropositivity

Characteristics such as gender, education level, BMI category, or personal history of diseases were not associated with changes in the probability of being positive for anti-SARS-CoV-2 antibodies (p > 0.05). The prevalence of anti-SARS-CoV-2 antibodies was lower in those aged ≥60 years (26.2%; 95% CI [20.7%–33.1%]). Other factors associated with seropositivity were living with 3 or more people, a family member with previous SARS-CoV-2 infection and have a deceased member in the household in the last five months (Table 3).

Table 3 Factors associated with positivity to SARS-CoV-2 in the general population in a high-altitude setting in Peru.

Models adjusted for age, gender, and study setting.

	Negatives 1284 (100%)	Positives 636 (100%)	Adjusted PR	
SELF-REPORTED COMORBIDITIES				
Hypertension	101 (7.85%)	27 (4.24%)	0.70 (0.48–1.02)	
Diabetes	64 (4.98%)	21 (3.30%)	0.87 (0.59–1.28)	
Asthma	23 (1.79%)	6 (0.94%)	0.63 (0.30–1.30)	
Cardiac disease	11 (0.87%)	1 (0.16%)	0.29 (0.05–1.96)	
Renal disease	12 (0.93%)	7 (1.10%)	1.10 (0.62–1.96)	
Cancer	9 (0.70%)	1 (0.16%)	0.29 (0.05–1.84)	
DEVELOPED SOME OF THESE SYMPTOMS IN THE LAST 3 MONTHS BEFORE THE SURVEY				
No symptoms	584 (45.5%)	188 (29.6%)	0.68 (0.58–0.81)	
Dysgeusia/Dysosmia	18 (1.4%)	118 (18.6%)	2.74 (2.41–3.12)	
Respiratory distress	23 (1.8%)	65 (10.2%)	2.18 (1.84–2.62)	
Fever	108 (8.4%)	138 (21.7%)	1.76 (1.52–2.05)	
General discomfort	136 (10.6%)	163 (25.6%)	1.75 (1.51–2.03)	
Cough	174 (13.5%)	178 (27.9%)	1.68 (1.48–1.92)	
Muscle pain	146 (11.4%)	149 (23.4%)	1.57 (1.35–1.82)	
Nasal congestion	105 (8.2%)	102 (16.0%)	1.47 (1.23–1.75)	
Back pain	216 (16.8%)	182 (28.6%)	1.46 (1.26–1.71)	
Diarrhea	96 (7.5%)	82 (12.9%)	1.40 (1.16–1.70)	
Headache	412 (32.0%)	275 (43.2%)	1.24 (1.08–1.44)	
Sore throat	272 (21.2%)	167 (26.2%)	1.11 (0.94–1.30)	
BEHAVIORS				
Always wear a face shield	338 (27.6%)	139 (23.0%)	0.73 (0.60–0.89)	
Always wear a face mask	1085 (85.8%)	511 (81.5%)	0.76 (0.63–0.92)	
Always use alcohol to hand disinfection	1039 (82.7%)	486 (77.5%)	0.78 (0.65–0.92)	
Always wash hands	1036 (83.7%)	484 (78.3%)	0.83 (0.69–0.99)	
Always stay more than 1 meter away from other people	945 (75.3%)	443 (70.8%)	0.85 (0.73–0.99)	
Always wear gloves	69 (5.9%)	31 (5.2%)	0.82 (0.59–1.12)	

Overall, 40.2% (95% CI [37.3%–43.4%]) of participants had no symptoms in the three months prior to the interview; however, 24.4% (95% CI [20.7%–28.6%]) of them were positive for anti-SARS-CoV-2 antibodies. Among those who developed any symptoms, 39.0% (95% CI [35.2%–43.2%]) were positive for anti-SARS-CoV-2 antibodies. Having no symptoms was associated with a 32% reduction in the probability of having anti-SARS-CoV-2 antibodies (aPR =0.68, 95% CI [0.58–0.81]); however, some symptoms increased the probability of having anti-SARS-CoV-2 antibodies, such as dysgeusia/dysosmia (aPR = 2.74, 95% CI [2.41−3.12]) and respiratory distress (aPR = 2.18, 95% CI [1.84–2.62]).

Finally, 1473 (92.2%) reported accomplishing with some protective behaviors; using at least one of them decreased the probability of having anti-SARS-CoV-2 antibodies by 32% (aPR = 0.68, 95% CI [0.55–0.87]). Specifically, always wearing a face shield decreased by 27% (aPR = 0.73, 95% CI [0.6–0.89]) the probability of being positive for anti-SARS-CoV-2 antibodies, whereas the use of face masks decreased it by 24% (aPR = 0.76, 95% CI [0.63–0.92]), and the use of alcohol to hand disinfection by 22% (aPR =0.78, 95% CI [0.65–0.92]).

Discussion

Principal findings

Our results indicate that, on average, a third of the population of Cusco had antibodies against SARS-CoV-2 virus, expanding our knowledge about the epidemiology of COVID-19 pandemic in high-altitude settings, which is a very high result compared to other seroprevalence studies in different settings and countries. These results may be appropriately contextualized to understand the varying spread and transmission of the epidemic (Eckerle & Meyer, 2020; Kucharski et al., 2020). In Peru, a mandatory social lockdown, suspension of tourism, and internal migration were quickly adopted after the detection of the first COVID-19 case (Presidencia del Consejo de Ministros, 2020); therefore, the first wave was presented relatively late compared to other countries in the region. Cusco reported the first positive case in March 2020; but it was not until June that a sustained increase in cases was identified, reaching the peak in August, and a decreased occurred since September (Dirección Regional de Salud de Cusco, 2020). Thus, our results were collected immediately after the end of the first wave in Cusco, and results should be compared with studies in a similar epidemiological situation.

Prevalence of anti-SARS-CoV-2 antibodies may differ depending upon the pandemic stage. A nationwide study in Spain reported a prevalence of 5% (Pollán et al., 2020) with markedly geographical variation (up to 10% in Madrid); however, a study in a region of Brazil, reported a population-based prevalence of 40% (Silva et al., 2020). Another study reported 16% in Chile, but was 24.5% in the Metropolitan region (Brault, 2021). Studies in other countries have not yet been officially published.

In Peru, some seroprevalence studies have already been reported; for example, during June–July 2020, the prevalence in Lima was estimated in 25.2% (Reyes-Vega et al., 2021), while in Cusco was 2.6% (Dirección Regional de Salud de Cusco, and Gobierno Regional Cusco, 2020). However, Lima had the peak of infections in June 2020 compared to August in Cusco. Other Peruvian cities have also reported high prevalence at the end of their first wave, such as Lambayeque (located on the northern coast of Peru and strongly affected) where the prevalence was 29.5% by June–July (Díaz-Vélez et al., 2021), and it was 71% in Iquitos (located in the Peruvian jungle), the highest reported in the country (Ministerio de salud, 2020). These studies used lateral immunochromatography tests (rapid tests) as a diagnostic method, which have obtained a diagnostic yield of less than 50% sensitivity in a field validation in Peru (Vidal-Anzardo et al., 2020). In the world, the tests used in prevalence studies are ELISA, chemiluminescence (CLIA), lateral flow chromatography (LFIA), or fluorescence immunoassays (FIA). In a systematic review of the validity of these tests, it indicates that the CLIA and ELISA tests for SARS-COV-2 have levels of sensitivity greater than 90%, while those of LFIA are in the range of 80 to 89% (Kontou et al., 2020). Although LFIA tests are more attractive for seroprevalence studies, prevalence adjustments must be made due to their low sensitivity. We used, however, CLIA tests, which in a validation against PCR-positive Peruvian samples for SARS-COV-2 and negative samples, stored since before the pandemic, showed a diagnostic yield greater than 95%; as a result, we consider that our estimates do not require other adjustments (Instituto Nacional de Salud of Peru. Lima, 2020).

In addition, our results show a marked difference in seroprevalence rates between cities in the same region. While, on average, we have a prevalence of 33%, this estimate in Cusco city almost doubled that of Quillabamba (38% vs. 20%). The national seroprevalence study, carried out in April 2020, in Spain (Pollán et al., 2020), has already shown that cities with >100,000 inhabitants have higher prevalence, probably due to more significant social interaction, as could happen in our study, where the Cusco city, with high demographic density, and the periphery of that city, were markedly affected. This has also been seen in other studies, where urban cities have a higher prevalence than those in rural locations, as in our study (Brault, 2021).

Some ecological studies evaluating contagion at high-altitudes cities reported that at higher altitudes were associated with lower frequency of the disease (Arias-Reyes et al., 2020; Cano-Pérez et al., 2020; Segovia-Juarez, Castagnetto & Gonzales, 2020; Thomson et al., 2021; Woolcott & Bergman, 2020), Contrary to these, the prevalence in our study was inverse to the altitude of the city, but potentially correlated with population density. Ecological design studies are important to propose hypotheses, but these have important biases (Aggarwal & Ranganathan, 2019; Gnaldi, Tomaselli & Forcina, 2018; Gordis, 2009; Lancaster, Green & Lane, 2006; Rousson, Rosselet & Paccaud, 2017). In the case of the COVID-19 pandemic, ecological studies are based on data from registration by surveillance systems that have great limitations (unidentified asymptomatic, different diagnostic criteria, passive registration, etc.), especially in cities located at high altitudes, which have less access to health services and a higher rate of poverty, so it is necessary to be very cautious when considering them. In our study, altitude does not appear to be a unique associated factor, though it will not be easily determined. Our study shows that in an urbanized city with a high population density, the prevalence is high regardless of the altitude at which they are located (Table 4). These factors would condition the prevalence of a disease (Huamaní et al., 2020); ecological studies do not usually include analyses of these confounding variables. Furthermore, ecological studies can hardly measure prevalence, especially when the disease has a high percentage of unidentified cases.

Table 4 Differences between cities and seropositivity to SARS-COV-2 antibodies in settings of Cusco.

	Cusco city	Periphery of Cusco	Quillabamba	
	(n = 640)	(n = 876)	(n = 408)	
% seropositive to SARS-CoV-2 antibodies	38.8% (33.4–44.9%)	34.9% (30.4–40.1%)	20.3% (16.2–25.6%)	
Altitude (m.a.s.l.)	3330	3330	1050	
Demographic density (aprox. inhabitants/km2)	1000	1100	10	
Access roads	Multiple land access routes, from within and outside the region. Air access from Lima (capital of Peru)	Only two land access routes, from the city of Cusco and nearby cities	
Temperature (°C) in September	7.9 [3.6–13.5]	14.6 [10.2 –19.2]	
Notes.

M.a.s.l. meters above the sea level

Factors associated with the seropositivity of SARS-CoV-2

Our results are consistent regarding some factors associated with a greater probability of positivity to anti-SARS-CoV-2 antibodies, for example, the symptoms described such as anosmia or respiratory distress, in the context of the pandemic, have a strong disease predictive association (Martin-Sanz et al., 2020). In recent systematic reviews and meta-analyses, anosmia has been similarly associated with anti-SARS-CoV-2 antibodies with an OR between 11 and 14 (Aziz et al., 2021; Pang et al., 2020). In our study, we used the prevalence ratio (PR) as an indicator because the prevalence of the disease was high (>20%) and the OR could overestimate the strength of the association. But when logistic regression analysis was conducted, the OR for dysgeusia/dysosmia, the OR was 14.27 (8.24–24.7), similar to that found in systematic reviews.

At the beginning of the pandemic, the first publications focused on contagion processes between contiguous people, such as clusters of families (Chan et al., 2020; Qian et al., 2020). These reports indicated that transmission between members of a family was not total, despite the existence of contact between the family members. The secondary attack rate among members of a family was around 36% (Grijalva et al., 2020), however, in population seroprevalence studies, intra-domiciliary transmission has not been studied. In our study, if one family member was positive, in 44% of the cases all family members were positive. This result may be more relevant for the purposes of adjusting the mathematical models of intra-household transmission.

Chronic diseases, such as hypertension, diabetes or obesity have already been described as predictors of poor prognosis of COVID-19 (Xu, Mao & Chen, 2020); therefore in Peru, the isolation of people with these diseases was recommended. Our study did not evidence that people with these antecedents had a different probability of contagion. In contrast, the older age groups (>60 years) do present a lower risk, perhaps due to the lower exposure as they do not have the same needs to leave home or because the other members of their family protected them. In Peru, only 12% of the population is older than 60 years old (in our study, it represents 25%), so the social behavior directed towards this vulnerable population may be protective and would explain the low prevalence in this group of age (Instituto Nacional de Estadistica e Informática-Peru, 2020), and as they are a population with low prevalence and high susceptibility, vaccination could show greater effectiveness than in the young population.

The actions of social distancing, wearing masks, and protection, in general, are included in several recommendations to reduce the spread of the infection, although there were discrepancies in their use (limit it only to symptomatic people, people at risk, among others) (Feng et al., 2020). The quantitative reduction value in contagion possibilities was evaluated in multiple studies, which resulted in a meta-analysis with positive results in favor of these measures (Chu et al., 2020). We identified a protective factor between 30% and 35% in practicing these measures, which leads us to continue recommending them. Although there is a decrease in risk, we could not identify some protective factors such as social distancing or hand washing, due to insufficient statistical power.

Since these measures are effective, inexpensive and easy to implement, it would be expected to be of general use. A study carried out in China during the pandemic early phase found that at least 84% of the population complied with some protection measure (Niu et al., 2020), in our study at the end of the first wave, it was observed that 92% of participants reported always followed some of the measures evaluated. Although the use was better accepted at the beginning of the pandemic, after the first wave, a decrease in use was expected, especially since Peru does not have a tradition in the widespread and constant use of masks or other protective measures.

Limitations

Our study is not representative for the whole Cusco region because of access to cities and distance among them was not easy in lockdown times. However, the representativeness of the cities with the largest population size and most affected was prioritized to facilitate decision-making in public health in the region, knowing that the prevalence is also lower in the cities with smaller population size.

Additionally, despite the indication of lockdown at the beginning of our study, the population had to work outside their homes; for this reason, the sample collection was mainly on Sundays to guarantee representativeness of the population.

In our study we used tests that detected total antibodies, and our results might not be easily compared with other studies that used tests that differences between IgM and IgG. However, the calculation of the prevalence of a disease is based on the identification of new and old cases, therefore, the development of one or another antibody allows them to be classified as “case”. Therefore, for the objectives of our study, there were no implications in the discrimination of IgM and IgG antibodies.

A high rejection rate may influence the characteristics of the data. Whereas rejection rate was not collected, it was not frequently reported by the supervisory team. Even so, individuals from Quillabamba rejected participation more than subjects from other settings.

Also, in Cusco there is little availability of confirmatory tests such as molecular tests, so we cannot confirm the diagnoses. However, the evaluation carried out at the National Institute of Health of Peru demonstrated a high reliability of the test of the test used in our study. Finally, lack of power can be an issue as some associations were not significant, although these showed a suggestive trend.

Conclusions

In conclusion, at the end of the first wave, the seroprevalence found in Cusco city, a high-altitude setting, was high, with significant differences between areas. The factors associated with a lower probability of having anti-SARS-CoV-2 antibodies have been widely recommended (wearing a mask, use of alcohol and hand washing), but efforts must be made to sustain them over time since there is still a high proportion of susceptible people.

Supplemental Information

Supplemental Information 1 Data

Click here for additional data file.

Supplemental Information 2 Individual questionnaire (translated from Spanish)

Click here for additional data file.

Supplemental Information 3 Extended analysis of Table 2

Click here for additional data file.

We would like to thank to the more than 40 professionals and medical students who participated in the execution of the study.

Additional Information and Declarations

Competing Interests

Author Contributions

Human Ethics

Data Availability

The authors declare there are no competing interests.

Charles Huamani conceived and designed the experiments, performed the experiments, analyzed the data, prepared figures and/or tables, authored or reviewed drafts of the paper, and approved the final draft.

Lucio Velásquez conceived and designed the experiments, performed the experiments, authored or reviewed drafts of the paper, and approved the final draft.

Sonia Montes and Ana Mayanga-Herrera performed the experiments, authored or reviewed drafts of the paper, and approved the final draft.

Antonio Bernabé-Ortiz conceived and designed the experiments, analyzed the data, prepared figures and/or tables, authored or reviewed drafts of the paper, and approved the final draft.

The following information was supplied relating to ethical approvals (i.e., approving body and any reference numbers):

The Ethics Committee of the Universidad Científica del Sur approved this study (code 051-2020-PRO99).

The following information was supplied regarding data availability:

The data are available in the Supplementary File.

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
