# Peer review of "SARS-CoV-2 seroprevalence in a high-altitude setting in Peru: adult population-based cross-sectional study"

_PeerJ, doi:10.7717/peerj.12149_

## Round 0.1 · original submission · Major Revisions

Your manuscript has scientific merit, after reading your work and the analysis of the experts. Please, take into account their comments so as to improve your paper in a revised version of the text.

·

Basic reporting

OK

Experimental design

OK

Validity of the findings

OK. Just attend to some concerns.

Additional comments

The authors analyzed the prevalence of antibodies to SARS-CoV-2 in Cusco, after the first epidemic wave. They found a prevalence of antibodies in all the populations groups over 30%, with positive correlation of antibody presence with household positivity and negative correlation with wearing mask all times. The information is relevant and interesting. Some minor concerns should be addressed for publication of this manuscript.

Major comments:
1. The authors described in Discussion differences in prevalence between countries and inside Peru, In addition to comparing the reported sensitivity of the tests used in each study, it would be interesting to mention the antigens used in each of these tests, and the limitations of comparing prevalence between different studies because of this heterogeneity.
2. In addition, although the companies argues for very high sensitivity and specificity, did the authors perform any quality control test to validate the performance of the test used: for example, sensitivity with sera from patents with known previous COVID-19 and specificity with sera collected before the pandemic?

Reviewer 2 ·

Basic reporting

The manuscript by Huamani et al. assessed the seroprevalence of SARS-COV-2 in a high-altitude setting in Peru. Overall, it is well written and the subject is interesting. I enjoyed reading it. The method section is clear and the results are significant. The discussion and conclusion are in accordance with the results.

Experimental design

The study is well designed and methods and analyses are clear.

Validity of the findings

The results are significant and presented properly. There is no comment.

Additional comments

In the introduction, it would be good that some information about the number of death and confirmed cases for Peru be added.

·

Basic reporting

This is an interesting study, following the trends of regional studies of SARS-CoV-2 seroprevalence in Peru.
Their findings highlight the importance of such information.

Experimental design

The experimental design is appropriate to address the research question.

Validity of the findings

Conclusions are well stated, linked to original research question.

Additional comments

This is an interesting study, following the trends of regional studies of SARS-CoV-2 seroprevalence in Peru.
Their findings highlight the importance of such information.
I only recommend to extend the Discussion regarding the implications of the findings coupled with the ongoing vaccination plan. Secondly, Figure 1, please add a relative map showing South America and Peru, in order to international audience have more clear the location of the study.
Please include the recent publications, even on on Peer J, of other similar studies in Peru, e.g. in Chiclayo or Lambayeque, at Lima, the capital in EClinical Medicine, among others.

Reviewer 4 ·

Basic reporting

In this study, Huamaní et al. estimated the seroprevalence of COVID-19 (?) in three regions of Cusco, Peru.
I found this study interesting and novel but there are many flaws that need to be addressed.

Experimental design

Please see below.

Validity of the findings

Please see below.

Additional comments

ABSTRACT:

It should indicate that the participants were ≥18 years old. The title should also indicate that the study included an adult population.

The statement “Little evidence exists about the prevalence of COVID-19 infection at high altitude.” is not accurate. The authors have not performed an exhaustive search of the literature. There are numerous studies that have estimated the prevalence of SARS-CoV-2 in high-altitude cities. This also applies to INTRODUCTION.

Based on the aim of the study (the estimation of the seroprevalence of COVID-19 in Cusco), this reviewer assumed that this study used a representative population of Cusco, but that does not seem to be the case. The aim of the study was not specifically addressed. Therefore, the conclusions of the study are misleading. Did the authors estimate the seroprevalence of COVID-19 and SARS-CoV-2? How was COVID-19 diagnosed?

Since the analysis of antibodies did not discriminate between IgM and IgG, then the authors should indicate that the diagnosis of SARS-CoV-2 infection was based on the identification of anti-SARS-CoV-2 total antibodies (IgM or IgG). More importantly, is there a valid justification for not performing a separate analysis for IgM and IgG? If data is available, this analysis should be performed and included in the study.


MATERIALS AND METHODS

It was mentioned that the cross-sectional study was part of a longitudinal cohort study. Which longitudinal study are the authors referring to? Has this longitudinal study already been published? If not, then further details of the main (longitudinal) study have to be provided.

At what temperature were the samples stored?

It was mentioned that up to 1800 participants were enrolled. However, Results show that 1924 participants were analyzed. This needs clarification.

Was there any difference in ethnicity across regions?

How was obesity diagnosed? Was it self-reported?

RESULTS:

The study aimed to estimate the seroprevalence of SARS-CoV-2 antibodies in Cusco, with emphasis on a high-altitude population. Since data were shown for 3 regions (one at low altitude), it is natural to expect a deeper analysis of the ORs for seroprevalence by altitude. Please provide the analysis of the OR for seroprevalence at Cusco City and periphery of Cusco vs. Quillabamba (reference), adjusted for age, gender, comorbidities, and population density.

Table 1: Are the values expressed as the media? Did you mean median or average? What do you mean by “Previous COVID-19 test”? Do data show only positive results? What kind of test? The statement “Asymptomatic during pandemic” is meaningless. Please specify the period/duration relative to the serological test performed in the study. Please change “Personal history” for “Self-reported medical history” or “Self-reported comorbidities”. Why information on other important comorbidities were not collected? In “personal history”, did you mean “obesity”? To be consistent, Tables 1 and 3 should show data for all comorbidities. Was type 2 diabetes actually discriminated from type 1? What were the diagnosis criteria used? Was information on gender (orientation) collected? Were the participants asked about their biological sex?

Table 2: Why does the title of the table refer to the prevalence of serum antibodies to SARS-CoV-2 in the general population in a high-altitude setting in Peru” if Quillabamba (~1000 m elevation) was included? Was not altitude defined (in the study) as an elevation over 2500 m? The comparison of prevalence across regions is interesting. Quillabamba had the lowest prevalence. This must be further discussed and contrasted with the findings from numerous previous studies (Quevedo-Ramirez, Respir Physiol Neurobiol 2020; Accinelli, Arch Bronconeumol 2020; Intimayta-Escalante, High Alt Med Biol 2020; Woolcott, High Alt Med Biol 2020; Seclen, Diabetes Res Clin Pract 2020, Perone Sci Total Environ 2021; Cano-Pérez, Am J Trop Med Hyg 2020.). These references should be included in Introduction. Please correct: ”Models adjusted by” (adjusted for…”). What do the authors mean by “adjusted for study setting”? Data cannot be adjusted for the same variable. This should be clarified in the table footnotes. Table should show the n size for each variable/category. Data show that only 50% of all who died had positive antibodies. Is this correct? This deserves extensive discussion. How was BMI measured?

The authors presented some of their data as adjusted odds ratios, however, all odds ratios were interpreted as probabilities. That is inappropriate and must be corrected.

Correct grammar and spelling throughout the manuscript. I would suggest the manuscript be revised by a native English speaker.

Table 3: “DEVELOPED SOME OF THESE SYMPTOMS IN THE LAST 3 MONTHS”. Three months prior to what date? HBP: does this acronym stand for high blood pressure? Did you mean hypertension? How was Cardiac disease and Renal disease defined? Could you explain in Methods? Could you use the proper terms for “Smell / taste alteration”? What do you mean by “Always wear…”. This does not sound accurate. I doubt the subjects used protective gear at home. Please clarify.

Figures for review should be provided using a common format. Not all have the applications to open EPS files. I could not see Figure 1.

DISCUSSION:

The fact that the first case in Cusco was reported in March 2020 and the blood samples collected in September 2020 may have introduced an important bias in the estimates of the prevalence considering that IgG and IgM were not separately analyzed. This is a major limitation of the study that should be discussed.

The comparison of prevalence across regions is interesting. Quillabamba had the lowest prevalence. This must be further discussed and contrasted with the findings from numerous previous studies (Quevedo-Ramirez, Respir Physiol Neurobiol 2020; Accinelli, Arch Bronconeumol 2020; Intimayta-Escalante, High Alt Med Biol 2020; Woolcott, High Alt Med Biol 2020; Seclen, Diabetes Res Clin Pract 2020, Perone Sci Total Environ 2021; Cano-Pérez, Am J Trop Med Hyg 2020.).

This study has numerous limitations, but I do not see any effort at self-criticism. It has been referred that this study is not representative of the whole population of Cusco. However, nothing is mentioned about the people who are hospitalized. This population may represent a large proportion of cases with seropositivity. Also, it should be discussed the rate of acceptance or rejection to be surveyed. Was there any difference on the participation among and between families across regions?

CONCLUSIONS:

The conclusion that “the altitude seems not to influence estimates” is not accurate. Your data suggest the opposite. I would like to see the unadjusted ORs for all your estimates in Table 2. In addition, as I suggested in my previous comment, the ORs estimates of seroprevalence for Cusco and peripheral regions vs Quillabamba should also be adjusted for comorbidities and population density, in addition to age and sex. Based on the additional findings from the analyses requested, discussion should be modified accordingly.

---

## Round 0.2 · Major Revisions

Major issues have been highlighted by one of the reviewers. Please, try to address their comments in a new revised version of the text.

From my point of view, I see the authors are indicating a rationale for not including ecological studies, which is correct. However, they are indicating "there are many ecological studies" without any references. If a reader wants to know more information about this point, they will be lost. Therefore, I consider that the authors should include some references at the end of the first phrase, including some ecological studies (not necessarily those suggested by the reviewer). Also, it is necessary to add references about the criticism to the ecological studies (methodological papers or books about epidemiology)

·

Basic reporting

Satisfactory

Experimental design

Ok

Validity of the findings

Valid

Additional comments

The authors addressed adequately the concerns of the reviewers.

Reviewer 4 ·

Basic reporting

Thank you for addressing most of my comments. Still, some points have not been properly addressed.

Experimental design

The work needs further revision.

Validity of the findings

The work needs further revision.

Additional comments

Given the limitations of the study acknowledged by the authors, and the presence of underadjustment (for instance, prevalence ratios were not adjusted for comorbidities) in a population-based study, the Conclusions of the study (Line 337) need revision. The statement "altitude seems not to influence estimates as the only variable, as done by other factors such as population density or population size" is not supported by the findings. The conclusions stated in the Abstract are supported by the main findings of the study. Thus, the Conclusions at the end of the manuscript should be consistent with those stated in the Abstract.

A significant part of the study focuses on the factors associated with SARS-CoV-2 seropositivity. The authors attempted to explain this association using regression analysis adjusting only for age, sex, and study setting. Thus, the limitation of undeardjustment must be acknowledged in the Discussion, more so when the authors argument to exclude many variables from their final model was based on the lack of significant correlation between the excluded variables and the dependent variable.

The authors are reluctant to discuss their findings in relation to those from numerous “ecological” studies published in the literature. These uncited studies have examined the prevalence of SARS-CoV-2 or COVID-19 in regions located at high altitude (Quevedo-Ramirez, Respir Physiol Neurobiol 2020; Accinelli, Arch Bronconeumol 2020; Intimayta-Escalante, High Alt Med Biol 2020; Woolcott, High Alt Med Biol 2020; Seclen, Diabetes Res Clin Pract 2020; Perone, Sci Total Environ 2021; Cano-Pérez, Am J Trop Med Hyg 2020). This is surprising and disappointing considering that their analysis included populations from three regions in Cusco, two located at high altitude and one at low altitude. The rationale for the analysis of these three Cusco regions only must be stated in Methods.

---

## Round 0.3 · accepted · Accept

All the reviewers' concerns have been correctly addressed.

Reviewer 4 ·

Basic reporting

Thank you for addressing all my comments.

Experimental design

no comment

Validity of the findings

no comment